# ProReason: Multi-Modal Proactive Reasoning with Decoupled Eyesight and Wisdom

## Abstract

Large vision-language models (LVLMs) have witnessed significant progress on visual understanding tasks. However, they often prioritize language knowledge over image information on visual reasoning tasks, incurring performance degradation. To tackle this issue, we first identify the drawbacks of existing solutions (*i.e.*, insufficient and irrelevant visual descriptions, and limited multi-modal capacities). We then decompose visual reasoning process into two stages: visual perception (*i.e.*, eyesight) and textual reasoning (*i.e.*, wisdom), and introduce a novel visual reasoning framework named ProReason. This framework features multi-run proactive perception and decoupled vision-reasoning capabilities. Briefly, given a multi-modal question, ProReason iterates proactive information collection and reasoning until the answer can be concluded with necessary and sufficient visual descriptions. Notably, the disassociation of capabilities allows seamless integration of existing large language models (LLMs) to compensate for the reasoning deficits of LVLMs. Our extensive experiments demonstrate that ProReason outperforms both existing multi-step reasoning frameworks and passive peer methods on a wide range of benchmarks for both open-source and closed-source models. In addition, with the assistance of LLMs, ProReason achieves a performance improvement of up to 15% on MMMU benchmark. Our insights into existing solutions and the decoupled perspective for feasible integration of LLMs illuminate future research on visual reasoning techniques, especially LLM-assisted ones.

## 1 Introduction

In recent years, large language models (LLMs) (Dubey et al., 2024; Team et al., 2023; Jiang et al., 2023) have experienced explosive growth in their capabilities, driving significant advancements across various fields (Shao et al., 2023; Guo et al., 2024; Shao et al., 2024). This progress has also sparked interest in developing large vision-language models (LVLMs) (Chen et al., 2024a; Bai et al., 2023), which, like LLaVA (Li et al., 2024b), have achieved remarkable performance in multi-modal understanding tasks. However, state-of-the-art (SOTA) LVLMs still struggle to integrate visual understanding with textual reasoning simultaneously due to inherent modality differences and deficient training data. For example, Ghosh et al. (2024) demonstrate that LVLMs often rely more on their prior language knowledge, neglecting visual information in multi-modal reasoning tasks such as visual chart understanding and math reasoning, resulting in performance degradation. Figure 2 illustrates a typical case of this issue, where the reasoning process remains irrelevant to the image.

To address the challenges, a promising solution is to extract visual information from images into textual form to assist LVLMs in reasoning (Yin et al., 2023b; Mitra et al., 2024). For example, Ghosh et al. (2024) instruct LVLMs to generate fine-grained captions to facilitate the subsequent reasoning process. However, these existing methods exhibit two primary limitations: (i) The visual extraction process is question-agnostic and reasoning-free, that the image description is not targeted for a given question, and reasoning is not involved to deduce extra information for better descriptions. This drawback, termed as "passive", results in irrelevant or insufficient information, and subsequent performance degradation. (ii) The reasoning process entangles visual understanding with textual reasoning abilities of a single LVLM. As mentioned above, the limited capabilities of LVLMs do not always successfully handle both abilities and produce high-quality reasoning. Hence, both of these limitations hinder the performance of LVLMs in multi-modal reasoning tasks.

To address both shortcomings, we propose a novel multi-step multi-modal reasoning framework named PROREASON, which features proactive (*i.e.*, question-oriented and reasoning-involved) visual extraction and disentangled vision-reasoning capabilities. Specifically, we first decouple multi-modal reasoning capacity into two sub-tasks: visual perception (*i.e.*, eyesight) and textual reasoning (*i.e.*, wisdom). The former focuses on understanding images and converting visual details into textual form, while the latter integrates the gathered information to draw final conclusions. As illustrated in Figure 1, our framework can be divided into three stages: `Action`, `Judgment`, and `Summary`, with each involving different agents. In the Action stage, a Dispatcher selectively engages a Vision Expert to capture additional visual information, or a Reasoning Expert to derive more insights. Unlike passive methods, all three agents operate based on the given question and current descriptions, effectively avoiding information redundancy or insufficiency. Subsequently, a Referee in the Judgment stage evaluates the sufficiency of known information, deciding whether to return to the Action stage or proceed to the Summary stage, where all information is consolidated for a Summarizer to generate the final answer. Notably, the disentangled vision-reasoning does not require vision-irrelevant roles (*i.e.*, Reasoning Expert, Referee, and Summarizer) to be performed by LVLMs, enabling the seamless integration of existing LLMs with proven strong reasoning abilities (Chang et al., 2024), thereby addressing the limitations of LVLMs.

Empirically, we evaluate PROREASON across four challenging visual reasoning benchmarks with both open-source and closed-source models. Extensive experiments demonstrate that PROREASON achieves significant and consistent performance improvement over both existing reasoning frameworks and state-of-the-art passive methods across all the benchmarks, with a peak enhancement as 13.2%. Besides, our comparative analysis highlights the necessity and relative importance of each sub-agent, as well as the advantages of proactive information acquisition. With the verified superiority over the simultaneous inherent usage of both capabilities, the decoupled perspective not only demonstrates that textual reasoning capabilities outweigh visual understanding for multi-modal reasoning tasks, but also exemplifies the feasibility of LLM-assisted LVLM reasoning, exhibiting performance improvements of up to 15% on the MMMU benchmarks (Yue et al., 2023).

The main contributions of this work are threefold:

- We propose a novel multi-modal reasoning framework named PROREASON, featuring iterative proactive perception and decoupled vision-reasoning capabilities.

- Empirically, we identify the drawbacks of existing solutions (*i.e.*, insufficient and irrelevant visual descriptions, and limited multi-modal capacities), and validate the successful mitigation of these issues with PROREASON by the superior performance over peer methods.

- PROREASON showcases the remarkable feasibility of integrating LLMs for boosted visual reasoning, illuminating the potential for LLM-assisted LVLM reasoning in future research.

## 2 PRELIMINARY OBSERVATIONS

We first analyze the behaviors of several existing methods on the MMMU (Yue et al., 2023) dataset, a challenging multi-modal benchmark requiring comprehensive college-level knowledge and fine-grained reasoning abilities. All experiments are conducted with three recent LVLMs for robustness: `Llama3-LLaVA-NeXT-8B` (Li et al., 2024a), `LLaVA-OneVision-Qwen2-7B-OV` (Li et al., 2024b) and `Qwen-VL-Chat` (Bai et al., 2023). Further details are provided in Sec. 4.1.

### 2.1 CHALLENGES IN VISUAL REASONING: LIMITATIONS OF LVLMS

Chain-of-Thought(CoT) (Wei et al., 2022) has been extensively verified to enhance the performance of LLMs. Here, we explore its impact on the reasoning performance of LVLMs. Similarly, we require the models to "think step by step" before generating final answers. As shown in Table 1, a counter-intuitive phenomenon is observed: compared to "Direct" answering method, **the introduction of CoT consistently incurs slight performance degradation across all three models.** Inspired by the findings of Ghosh et al. (2024) that LVLMs often rely on prior language knowledge yet neglecting visual information, we hypothesize that the degraded performance might be caused by the image-irrelevant reasoning of CoT.

Table 1: Performance of three recent LVLMs on MMMU dataset with different assisting techniques.

| Method | Model | | |
|--------|-------|---|---|
| | Llama3-LLaVA-NeXT-8B | LLaVA-OneVision-Qwen2-7B-OV | Qwen-VL-Chat |
| Direct | 41.8 | 49.1 | 36.2 |
| CoT | 41.5 | 46.5 | 35.4 |
| VDGD | 42.3 | 49.7 | 37.1 |

Table 2: Image Relevance Score of Chain-of-Thought reasoning traces for "True" and "False" responses of three LVLMs, respectively.

| Model | Image Relevance Score | |
|-------|:---:|:---:|
| | True | False |
| Llama3-LLaVA-NeXT-8B | 3.70 | 3.36 |
| LLaVA-OneVision-Qwen2-7B-OV | 4.27 | 3.67 |
| Qwen-VL-Chat | 3.63 | 2.80 |

Table 3: Effectiveness evaluation of passive captions along Detail Level, Question Relevance, and Reasoning Effective Info Inclusion. "True" and "False" denote the response correctness of Llama3-LLaVA-NeXT-8B.

| Score | Llama3-LLaVA-NeXT-8B | |
|-------|:---:|:---:|
| | True | False |
| Detail Level | 4.43 | 3.93 |
| Question Relevance | 3.87 | 3.30 |
| Reasoning Effective Info Inclusion | 3.91 | 3.57 |

To validate the hypothesis, we design a pipeline referencing Liu et al. (2023c) to measure the correlation between model responses and image contents. Specifically, given a question, we take the reasoning process generated by GPT-4o-mini[1], whose superiority on MMMU task is verified in Sec. 4.2, as the golden solution. Subsequently, a more powerful judge (*i.e.*, GPT-4 (Achiam et al., 2023)) is employed to evaluate the relevance of responses of candidate models (e.g., Llama3-LLaVA-NeXT-8B) and this golden solution with the prompt template in Figure 8, and gives a comprehensive score from 1 to 5, with 5 indicating the highest relevance. Since the golden solution contains the essential information needed to solve the problem within the image, a higher correlation indicates a stronger relevance to the image content. As shown in Table 2, the reasoning chains of incorrect responses are significantly less relevant to the images than those of correct answers. An exemplary case is shown in Figure 2, where Llama3-LLaVA-NeXT-8B attempts to use the right-hand rule to solve the task, but its reasoning process is almost unrelated to the image, ultimately leading to an incorrect conclusion. This highlights the limitation of CoT for multi-modal reasoning tasks, that **image-irrelevant reasoning process provokes performance degradation, since the limited capabilities of LVLMs cannot effectively handle both image and text information simultaneously.**

## 2.2 DRAWBACKS OF PASSIVE INFORMATION EXTRACTION

**Passive visual reasoning techniques suffer insufficient and irrelevant visual information**, despite mitigating the oversight of images by converting them into detailed captions, like Visual Description Grounded Decoding (VDGD) (Ghosh et al., 2024). To support this claim, we generate fine-grained image captions using GPT-4o-mini with the prompt shown in Figure 9. We then incorporate these captions into the prompts for LVLMs to facilitate the reasoning process. As listed in Table 1, while these image descriptions improve the performance of LVLMs, the gains are marginal,

---

[1] https://openai.com/index/gpt-4o-mini-advancing-cost-efficient-intelligence

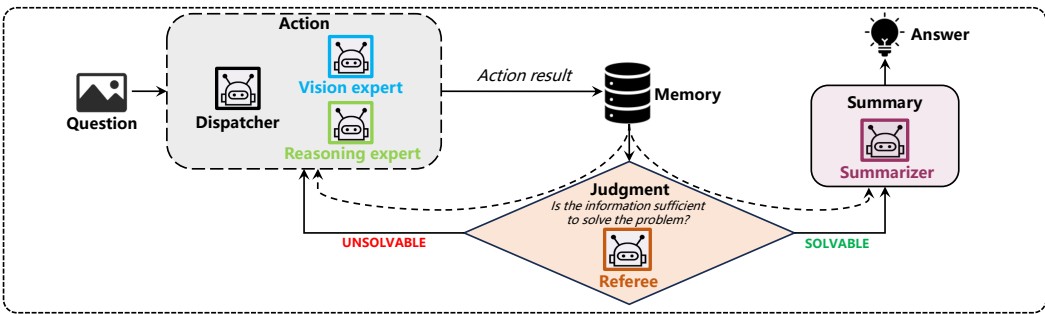

(1) PROREASON

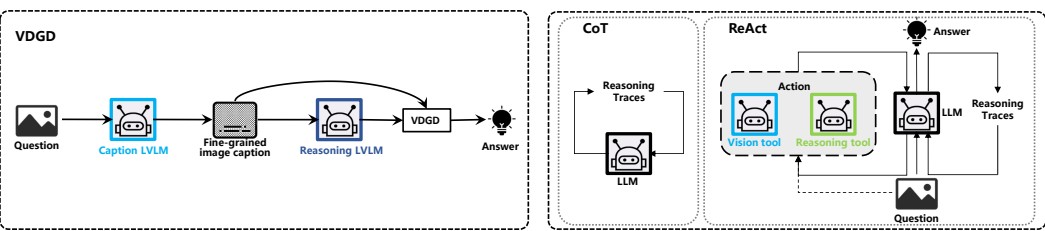

(2) Passive Visual Reasoning Enhancement Technique

(3) Multi-step Reasoning Framework for LLMs

Figure 1: Overview and comparison of workflows of PROREASON, VDGD and ReAct. Unlike existing works (e.g., VDGD and ReAct), our proposed method enables the model to actively acquire necessary information from the images, making it more suitable for LVLMs.

consistently amounting to less than 1%. This underscores the limited utility of captions generated by passive methods.

For further demonstration, we analyze the generated captions along three dimensions: Detail Level, Question Relevance, and Reasoning Effective Info Inclusion, measuring the richness of detail, relevance to the given question, and the inclusion of information necessary for reasoning, respectively. Similar to Sec. 2.1, we pair the image captions with the reasoning process of `GPT-4o-mini`, and employ `GPT-4` as a judge to rate the captions from 1 to 5 across these dimensions. The prompt template is shown in Figure 8. As shown in Table 3, the captions for correct responses of `Llama3-LLaVA-NeXT-8B` receive higher scores across all three criteria, highlighting the importance of better captions for multi-modal reasoning. Additionally, all captions score significantly lower in the Question Relevance and Reasoning Effective Info Inclusion dimensions than the Detail Level dimension, indicating that while **the captions are detailed, they often lack relevance to the questions.** Figure 2 shows a case where `Llama3-LLaVA-NeXT-8B` utilizes fine-grained captions to solve a question from the MMMU benchmark. As illustrated, although the caption exhaustively describes the image content, it incorrectly describes the wires in the image as octagons, and misses information about the locations of these wires. This information is irrelevant to the target question, thus offering minimal assistance to LVLMs. In summary, our analysis highlights the drawbacks of passive visual reasoning enhancement techniques in terms of information insufficiency and redundancy, due to their question-agnostic property.

## 3 METHOD

As illustrated in Figure 1, PROREASON consists of five functionally distinct yet inter-cooperative sub-agents, along with a Memory component. The entire workflow comprises three steps: Action, Judgment, and Summary, effectively integrating the proactive visual understanding (*i.e.*, eyesight) and textual reasoning (*i.e.*, wisdom) capacities for enhanced multi-modal reasoning performance.

## 3.1 ACTION STEP

Action step is the core of question-oriented visual information extraction, driven by three sub-agents: Dispatcher, Vision Expert, and Reasoning Expert. The Dispatcher orchestrates the workflow, selectively directing the Vision Expert to capture specific visual information, or instructing the Reasoning Expert to analyze known information to derive more. The responses from both the Vision Expert and the Reasoning Expert are stored in a textual Memory component.

Formally, given an image $I$ and its corresponding textual question $Q$, the Dispatcher decides to consult the Vision Expert or Reasoning Expert, based on the analysis of $Q$ and the known information in the Memory (if not empty). The Dispatcher then generates a query $q$ for the chosen expert. If the Vision Expert is selected, it takes the image $I$ and query $q$ as input, and generates an answer $A_v$, which is then stored in the Memory. When the Reasoning Expert is selected, it provides a response consisting of the reasoning process and the final answer $A_r$ based on the query $q$ and known information in the Memory, before only $A_r$ is stored in Memory. Notably, the Memory component allows PROREASON to keep compact information, and avoids lengthy reasoning traces like CoT and ReAct, thereby suffering less from redundant information.

## 3.2 JUDGMENT STEP

Judgment step is conducted by a sub-agent called Referee, which evaluates whether the information stored in Memory is sufficient to answer the question $Q$. The input to the Referee includes the question $Q$ and available information in the Memory. If the Memory contains adequate information to answer the question $Q$, the Referee outputs the identifier "SOLVABLE"; otherwise, it outputs "UNSOLVABLE". If the Referee's output is SOLVABLE, the workflow precedes to the Summary phase. Conversely, if the output is UNSOLVABLE, the Action phase is re-executed to gather more necessary information. This allows the Referee to collaborate closely with the three sub-agents in the Action step, enabling the framework to proactively acquire the necessary information and prevent omissions, thereby overcoming the drawbacks of passive methods.

## 3.3 SUMMARY STEP

The Summary step focuses on integrating the available information in the Memory, and providing the final answer to the question $Q$. This step is mainly powered by a sub-agent called Summarizer. Once the Referee determines that the information in the Memory is sufficient to address the question $Q$, and outputs the identifier SOLVABLE, the Summarizer will be called to draw a conclusion, based on the Memory information. This conclusion represents the answer to the question $Q$ of the whole framework, and will be evaluated by the performance metrics.

## 3.4 ADVANTAGES OF PROREASON

**Reduced information Mission or Redundancy.** With the close collaboration of the Dispatcher, Vision Expert, Reasoning Expert, and Referee agents, PROREASON can proactively (*i.e.*, question-orientedly and reasoning-involvedly) extract the necessary and sufficient visual details from images, effectively avoiding information omission or redundancy. Meanwhile, the Memory component retains only the image captions from the Vision Expert and the reasoning results from the Reasoning Expert, providing a compact textual descriptions. This contributes to minimize the interference of irrelevant information on the subsequent Summarizer.

**LLM-assisted Multi-modal Reasoning.** In PROREASON, the complete multi-modal reasoning process is decomposed into visual perception and textual reasoning stages, each executed by separate agents. These agents are then effectively organized through a designate pipeline to complete the tasks. This decomposition-before-integration approach not only circumvents the inherent differences between modalities, but also allows visually irrelevant sub-agents to be performed by text-only LLMs. Consequently, the extensively verified powerful reasoning capabilities of LLMs can be seamlessly synergized to power multi-modal reasoning process for performance enhancement.

## 4 EXPERIMENTS

In this section, we first evaluate the performance of our PROREASON framework against recent base-lines on multiple benchmarks, followed by an in-depth ablation analysis of different components.

### 4.1 GENERAL SETUP

**Datasets.** To comprehensively validate the performance of our framework, we conduct experi-ments across four benchmarks: Multi-modal Large Language Model Evaluation (MME) (Yin et al., 2023a), Massive Multi-discipline Multi-modal Understanding and Reasoning (MMMU) (Yue et al., 2023), MathVista (Wang et al., 2024), and HallusionBench (Liu et al., 2023a). All of them require visual reasoning capabilities to complete the tasks correctly, and are introduced briefly as follows:

- **MME** is an inclusive benchmark that encompasses 14 subtasks, designed to evaluate per-ceptual and cognitive abilities. Due to our emphasis on visual reasoning, we select the cognition-relevant tasks, including Commonsense Reasoning, Numerical Calculation, Text Translation, and Code Reasoning.[2]

- **MMMU** evaluates multi-modal models with multidisciplinary tasks that require college-level domain-specific knowledge and detailed reasoning. It comprises 11,500 questions across 30 disciplines and 183 sub-fields, emphasizing advanced perception and domain-specific reasoning.

- **MathVista** focuses on more challenging mathematical reasoning tasks that demand precise visual recognition and compositional reasoning. It includes 6,141 examples from 31 multi-modal mathematics datasets.

- **HallusionBench** evaluates models' ability to reason with images such as statistical charts, emphasizing nuanced visual understanding. It consists of 346 images paired with 1,129 questions, meticulously crafted by experts.

**Base Models.** We employ `GPT-4o-mini`, `Llama3-LLaVA-NeXT-8B` and `Qwen2.5-72B-Instruct` (Team, 2024) as our base models, due to their excellent performance, accessibility, and representativeness of model sources. As one of the most performant LVLMs, `GPT-4o-mini` demonstrates significant advancements in visual reasoning capabilities, and provides cheap and fast API. In contrast, `Llama3-LLaVA-NeXT-8B` is a fully open-sourced LVLM developed by the LLaVA team based on the Llama-3-8B LLM (Dubey et al., 2024) and CLIP vision encoder (Rad-ford et al., 2021). `Qwen2.5-72B-Instruct` is a robust LLM designed to deliver high-quality instruction following and handle complex tasks. It will be utilized for achieving LLM-assisted multi-modal reasoning.

**Baselines.** Besides the most basic method where models are instructed to answer questions di-rectly, we compare PROREASON with two categories of peer methods. To determine the benefits of proactive information extraction in PROREASON, we first consider two SOTA passive visual reason-ing methods, VDGD (Ghosh et al., 2024) and CCoT (Mitra et al., 2024). Additionally, we choose two multi-step reasoning frameworks of LLMs, Chain of Thought (CoT) (Wei et al., 2022) and ReAct (Yao et al., 2022), to demonstrate the effects of directly migrating LLM solutions to LVLMs.

- **Direct.** As indicated by the name, models are required to answer questions directly without dedicated prompts. This baseline is set to evaluate the initial performance of base models.

- **VDGD.** With image caption prefixed to text instruction, VDGD selects tokens that deviate the least from the description using a formula based on Kullback-Leibler divergence.

- **CCoT.** Given an image and the question, CCOT first generates a scene graph of the image with LVLMs, and then extracts the answer by prompting the LVLMs with the graph.

- **CoT.** CoT is an advanced prompting method that encourages LLMs to break complex tasks down into a series of easy steps, which has been applied broadly and verified to boost the reasoning performance remarkably (Chu et al., 2023).

---

[2]To facilitate the comparison across different benchmarks, the results for MME benchmark are calculated by the percentage of correct answers out of the total answers. For the results using the official MME calculation method, please refer to Table 9, Table 10, Table 11, and Table 12.

Table 4: Performance of multiple approaches with two base models across four visual reasoning benchmarks. "Hallu." is the abbreviation of HallusionBench. Based on the performance of the direct method, red and blue signify the improvement and degradation, respectively.

| Model | Method | Dataset | | | |
|---|---|---|---|---|---|
| | | MME | MMMU | MathVista | Hallu. |
| Llama3-LLaVA-NeXT-8B | Direct | 61.5 | 41.8 | 37.1 | 45.8 |
| | VDGD | 68.8 (+7.3) | 42.3 (+0.5) | 36.1 (-1.0) | 44.2 (-1.6) |
| | CCoT | 68.9 (+7.4) | 40.5 (-1.3) | 36.8 (-0.3) | 37.4 (-8.4) |
| | CoT | 58.8 (-2.7) | 41.5 (-0.3) | 35.9 (-1.2) | 43.1 (-2.7) |
| | ReAct | 68.5 (+7.0) | 46.7 (+4.9) | 31.7 (-5.4) | 43.6 (-2.2) |
| | PROREASON | 71.5 (+10.0) | 52.5 (+10.7) | 38.8 (+1.7) | 50.9 (+5.1) |
| | **Average** | 66.30 | 44.22 | 36.06 | 44.16 |
| GPT-4o-mini | Direct | 79.2 | 48.4 | 53.0 | 56.0 |
| | VDGD | 82.3 (+3.1) | 51.4 (+3.0) | 51.2 (-1.8) | 52.4 (-3.6) |
| | CCoT | 80.8 (+1.6) | 54.2 (+5.8) | 53.6 (+0.6) | 56.7 (+0.7) |
| | CoT | 87.8 (+8.6) | 58.5 (+10.1) | 53.8 (+0.8) | 56.3 (+0.3) |
| | ReAct | 87.3 (+8.1) | 54.8 (+6.4) | 49.3 (-3.7) | 51.1 (-4.9) |
| | PROREASON | 90.4 (+11.2) | 61.6 (+13.2) | 54.9 (+1.9) | 58.9 (+2.9) |
| | **Average** | 84.63 | 54.82 | 52.63 | 55.23 |

Table 5: Performance of Llama3-LLaVA-NeXT-8B Vision Expert, assisted by different text-only LLMs, on various benchmarks under ReAct and PROREASON frameworks. The red and blue texts indicate the improvements and reductions relative to the case of using the same method without the assistance of text-only LLMs, respectively.

| Vision Expert | Textual Sub-Agents | Method | Dataset | | | |
|---|---|---|---|---|---|---|
| | | | MME | MMMU | MathVista | Hallu. |
| Llama3-LLaVA-NeXT-8B | Llama3-LLaVA-NeXT-8B | Direct | 61.5 | 41.8 | 37.1 | 45.8 |
| | | ReAct | 68.5 | 46.7 | 31.7 | 43.6 |
| | | PROREASON | 71.5 | 52.5 | 38.8 | 50.9 |
| | Qwen2.5-72B-Instruct | ReAct | 71.0 (+2.5) | 50.4 (+3.7) | 34.6 (+2.9) | 40.4 (-3.2) |
| | | PROREASON | 81.3 (+9.8) | 56.8 (+4.3) | 48.8 (+10.0) | 52.3 (+1.4) |
| | GPT-4o-mini | ReAct | 73.6 (+5.1) | 48.4 (+1.7) | 36.2 (+4.5) | 46.7 (+3.1) |
| | | PROREASON | 84.7 (+13.2) | 54.5 (+2.0) | 41.7 (+2.9) | 53.1 (+2.2) |

- **ReAct.** ReAct is an LLM-specific agent framework, which performs tasks by alternating between reasoning and execution behaviors. To extend it to multi-modal domain, we use two LVLMs to perform both steps, and rename them as the Vision and Reasoning Experts, respectively. This aligns with our notions for easy understanding, and is shown in Figure 1.

**Implementation Details.** The prompt templates for all methods are shown in Figures 9, 10 and 11. Specifically for PROREASON, in order to prevent an infinite loop, if Dispatcher selects Vision Expert or Reasoning Expert to obtain information up to 5 times, and Referee still judges that the existing knowledge in Memory is not enough to solve the question, the information collection phase will be terminated immediately, and then Summarizer will directly answer the question based on the available information in Memory. This setup, refined through multiple trials, is the most effective. Additionally, since we cannot obtain the tokens output by GPT-4o-mini, we omit the step of selecting the token with the smallest deviation from the image description when implementing VDGD for GPT-4o-mini.

## 4.2 MAIN RESULTS

**PROREASON exhibits significant and consistent performance enhancement over baselines across all the benchmarks.** As listed in Table 4, despite better performance than the direct method on MME dataset, VDGD and CCoT fail to demonstrate consistent improvements on the other datasets. In contrast, PROREASON consistently outperforms all the other baselines for both `Llama3-LLaVA-NeXT-8B` and `GPT-4o-mini` model across all benchmarks, with a peak improvement as 13.2%, demonstrating the superiority and task robustness of PROREASON.

**Proactive information acquisition surpasses SOTA passive methods**, especially in complex visual reasoning tasks. Specifically, compared to MME, MathVista and HallusionBench present higher image complexity and question difficulty, and thus require stronger visual understanding and textual reasoning capabilities. This leads to performance degradation of passive methods (*i.e.*, VDGD and CCoT), highlighting their limited applicability to complex visual reasoning tasks. In contrast, PROREASON achieves notable performance improvements, up to 5.1%, by proactively acquiring visual information from images rather than generating question-agnostic captions. This aligns with out previous obervations in Sec. 2.2 that passive methods introduce substantial information redundancy or omission, misleading subsequent reasoning processes.

**Decoupling the visual perception and textual reasoning capabilities of an LVLM outperforms their simultaneous inherent usage.** As listed in Table 4, when both capabilities are utilized concurrently, CoT consistently degrades performance compared to the "Direct" method across all benchmarks with `Llama3-LLaVA-NeXT-8B` model, consistent with the findings in Sec. 2.1. In contrast, despite the same models, PROREASON alternates between visual information acquisition and textual reasoning processes, allowing to leverage each capability more effectively. This enables PROREASON to consistently outperform CoT with both `Llama3-LLaVA-NeXT-8B` and `GPT-4o-mini` across all benchmarks, demonstrating the effectiveness of capability decoupling.

**PROREASON outperforms ReAct with even less token consumption.** Given the similarity in multi-step reasoning, we compare PROREASON and ReAct in terms of performance and cost. Specifically, as an LLM-specific multi-step reasoning framework, ReAct only outperforms the "Direct" method on MME and MMMU, but underperforms on MathVista and HallusionBench, showing inferior performance compared to the consistent improvements of PROREASON. Furthermore, we compare their average token consumption of `GPT-4o-mini`. As shown in Table 8 of Appendix A, PROREASON consumes significantly fewer tokens than ReAct on both MME and MathVista tasks, indicating its higher token efficiency and the importance of a compact Memory in reducing token usage. Coupled with better performance, this suggests the superiority of PROREASON over LLM-specific ReAct framework.

**Text-only LLMs can be effectively integrated into PROREASON for dramatically enhanced performance.** As mentioned in Sec. 3.4, the decoupled visual perception and textual reasoning capabilities facilitate the seamless integration of text-only LLMs. To demonstrate the utility of this advantage, we fix the Vision Expert as `Llama3-LLaVA-NeXT-8B`, and replace other agents with text-only LLMs. As listed in Table 5, with the assistance of either `Qwen2.5-72B-Instruct` or `GPT-4o-mini`, the `Llama3-LLaVA-NeXT-8B` Vision Expert receives remarkable performance boost across all benchmarks, particularly by 15% on MMMU and 11.7% on MathVista, compared to directly providing answers. In contrast, ReAct gains a much smaller improvement. This highlights the unique advantage of PROREASON in leveraging existing text-only LLMs for enhanced performance. Notably, this advantage may open new avenues for continuously pushing the performance limits of LVLMs with the assistance of existing powerful LLMs.

## 4.3 RELATIVE IMPORTANCE OF SUB-AGENTS

To assess the importance of each sub-agent within the PROREASON framework for visual reasoning tasks, we design five scenarios where `Llama3-LLaVA-NeXT-8B` acts as Dispatcher, Vision Expert, Reasoning Expert, Referee, or Summarizer, respectively, while the other sub-agents are powered by `GPT-4o-mini`. Given that `Llama3-LaVA-NeXT-8B` exhibits weaker visual understanding and textual reasoning capabilities than `GPT-4o-mini`, the more significant the performance drop incurred by replacing a sub-agent with `Llama3-LaVA-NeXT-8B` is, the more important that sub-agent is. Here we primarily consider the MME and MMMU benchmarks due to their comprehensive question coverage. The experimental results are presented in Table 6.

Table 6: Performance of PROREASON across five scenarios for sub-agent assessment on visual reasoning tasks. For each scenario, one sub-agent is replacing with `Llama3-LLaVA-NeXT-8B`, while the others are performed by `GPT-4o-mini`. The blue text indicates the performance decline compared to the scenario with all agents performed by `GPT-4o-mini`.

| Dataset | GPT-4o-mini | Llama3-LLaVA-NeXT-8B | | | | |
|---------|-------------|-----------|---------------|------------------|------------|------------|
| | | Dispatcher | Vision Expert | Reasoning Expert | Referee | Summarizer |
| MME | 90.4 | 88.8 (-1.6) | 84.7 (-5.7) | 88.7 (-1.7) | 89.6 (-1.1) | 84.2 (-6.2) |
| MMMU | 61.6 | 60.9 (-0.7) | 54.5 (-7.1) | 60.2 (-1.4) | 51.5 (-10.1) | 51.0 (-10.6) |

Table 7: Performance of PROREASON with different configurations for the relative importance assessment between visual understanding and textual reasoning capabilities on visual reasoning tasks. The red text highlights the performance improvements brought about by the introduction of `GPT-4o-mini`.

| Dataset | GPT-4o-mini | | Llama3-LLaVA-NeXT-8B | | |
|---------|-------------------|---------------|------------------|------|--------|
| | Textual Sub-Agents | Vision Expert | All Sub-Agents | COT | Direct |
| MME | 84.7 (+13.2) | 77.8 (+6.3) | 71.5 | 58.8 | 61.5 |
| MMMU | 54.5 (+2.0) | 53.4 (+0.9) | 52.5 | 41.5 | 41.8 |

**Summarizer is the most crucial sub-agent, closely followed by Referee.** The replacement of Summarizer results in the most notable performance decline on both MME and MMMU tasks, reaching 6.2% and 10.6%, respectively. This highlights the critical function of the Summarizer in integrating all available information to conclude final answers. Besides, the substitution of Referee leads to a 10.1% reduction on MMMU. Given that MMMU is more challenging than MME, this finding underscores the essential role of the Referee in assessing the sufficiency of information, particularly in more complex visual reasoning tasks.

**Relatively, Dispatcher and Reasoning Expert are the least essential sub-agents.** Specifically, despite a decline, these two sub-agents exhibit significantly less performance degradation than other sub-agents. This can be attributed to the easier task of the Dispatcher, which requires minimal textual reasoning capabilities, and the infrequent calls of the Reasoning Expert, which is only activated when additional information needs to be inferred—a situation that is rare in current benchmarks. Besides, both sub-agents operate within the acquisition loop, allowing for greater error tolerance. Even if some error occurs, subsequent iterations can compensate for the missing information.

In summary, each sub-agent contributes to the performance of PROREASON, underscoring their necessity. Relatively, the Summarizer and Referee are the most critical sub-agents, while the Dispatcher and Reasoning Expert have the least impact.

## 4.4 WHICH ONE IS MORE CRUCIAL: VISUAL UNDERSTANDING OR TEXTUAL REASONING?

PROREASON effectively decouples the visual understanding and textual reasoning capabilities of LVLMs. However, it remains unclear which of these two capacities is more critical for visual reasoning tasks. To answer this question, we conduct comparative experiments of the following three scenarios:

- **`Llama3-LLaVA-NeXT-8B` as All Sub-Agents**. All sub-agents within PROREASON framework are performed by `Llama3-LLaVA-NeXT-8B` model.
- **`GPT-4o-mini` as Vision Expert**. Based on the above scenario, we implement the Vision Expert with `GPT-4o-mini`, while keep the other textual sub-agents unchanged.
- **`GPT-4o-mini` as Textual Sub-Agents.** Reversely, we utilize `Llama3-LLaVA-NeXT-8B` as the Vision Expert, and `GPT-4o-mini` for the other vision-irrelevant sub-agents.

**Textual reasoning capabilities outweigh visual understanding for multi-modal reasoning tasks, although both are important.** As shown in Table 7, replacing either the Vision Expert or the other agents with the more capable `GPT-4o-mini` achieves consistent performance enhancement,

highlighting the significance of both capabilities. However, substituting the textual sub-agents with `GPT-4o-mini` results in a more substantial performance boost compared to replacing the Vision Expert. This underscores the greater importance of textual reasoning over visual understanding for multimodal reasoning tasks, aligning with our previous analysis in Sec. 4.3 that identifies the Summarizer and Referee as the most crucial sub-agents.

## 5 RELATED WORK

**Large Visual-Language Model.** Recently, large vision-language models (LVLMs) (Bai et al., 2023; Chen et al., 2023; Liu et al., 2024b) have garnered widespread attention and demonstrated remarkable advancements in understanding and generating multi-modal contents. In the open-source domain, numerous LVLMs, like LLaVA (Liu et al., 2023c;b; 2024a; Li et al., 2024a;b) and InternVL (Chen et al., 2024b) families, have been extensively developed. In the closed-source domain, proprietary models such as GPT-4 (Achiam et al., 2023) and Gemini Pro 1.5 (Reid et al., 2024) have also achieved significantly success. Despite these advancements, existing LVLMs still encounter challenges in effectively integrating visual understanding with textual reasoning capabilities simultaneously. This limitation is particularly evident in their diminished attention to image content during visual reasoning process, such as chart interpretation and visual math reasoning, leading to degraded performance (Ghosh et al., 2024) and motivating more effective solutions.

**Passive Visual Reasoning.** Extracting information from images into text can effectively assist LVLM in performing visual reasoning tasks. Visual Description Grounded Decoding (VDGD) (Ghosh et al., 2024) first describes the image before prefixing this description to the prompt, assisting LVLMs on visual reasoning tasks. Furthermore, Compositional Chain-of-Thought (CCoT) (Mitra et al., 2024) directs LVLMs to generate scene graphs (SGs) that serve as a bridge between the visual and textual domains, aiding LVLMs in subsequent tasks. However, most of these methods employ a question-agnostic and reasoning-free visual extraction process, where image descriptions are not tailored to specific questions, and no reasoning is applied to infer additional information for improved descriptions. These "passive" approaches lead to the inclusion of irrelevant or redundant information, ultimately degrading performance. In contrast, PROREASON adopts question-oriented agents to collect necessary and sufficient information, effectively circumventing these drawbacks.

**Multi-step Reasoning Framework.** Multi-step reasoning frameworks have been developed for LLMs to achieve better performance by breaking down complex questions into easier ones (Pan et al., 2024). As a representative method, Chain-of-Thought (CoT) (Wei et al., 2022) enhances the arithmetic and commonsense reasoning capabilities by explicitly generating intermediate reasoning steps before concluding the final answers. Tree-of-Thoughts (ToT) (Yao et al., 2024) further refines the CoT mechanism by allowing LLMs to consider multiple reasoning paths and do self-assessment before making decisions. Considering that the inherent knowledge of LLMs may not be sufficient to complete tasks, ReAct (Yao et al., 2022) integrates information retrieval into the reasoning chains, enabling models to pause to verify results and determine whether additional information is needed before proceeding. Nevertheless, multi-step reasoning frameworks designed for text-only LLMs are not fully applicable to visual reasoning tasks, and may even impair the performance of LVLMs (Ghosh et al., 2024).

## 6 CONCLUSION

In this paper, we first validate that existing multi-modal reasoning approaches still suffer insufficient and irrelevant visual descriptions, as well as limited multi-modal capacities. To address these issues, we decompose the visual reasoning process into visual perception and textual reasoning stages, and introduce a novel visual reasoning framework named PROREASON, featuring multi-run proactive perception and decoupled vision-reasoning capabilities. Empirically, extensive experiments demonstrate the superiority of PROREASON over both passive image information acquisition methods and multi-step reasoning frameworks for text-only LLMs across multiple visual reasoning benchmarks with both open-source and closed-source models. Notably, our method showcases the remarkable feasibility of integrating LLMs for multi-modal reasoning with dramatically improved performance, highlighting the great potential for LLM-assisted LVLM reasoning in future research.

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

# A APPENDIX

## A.1 DEMONSTRATIVE EXAMPLES

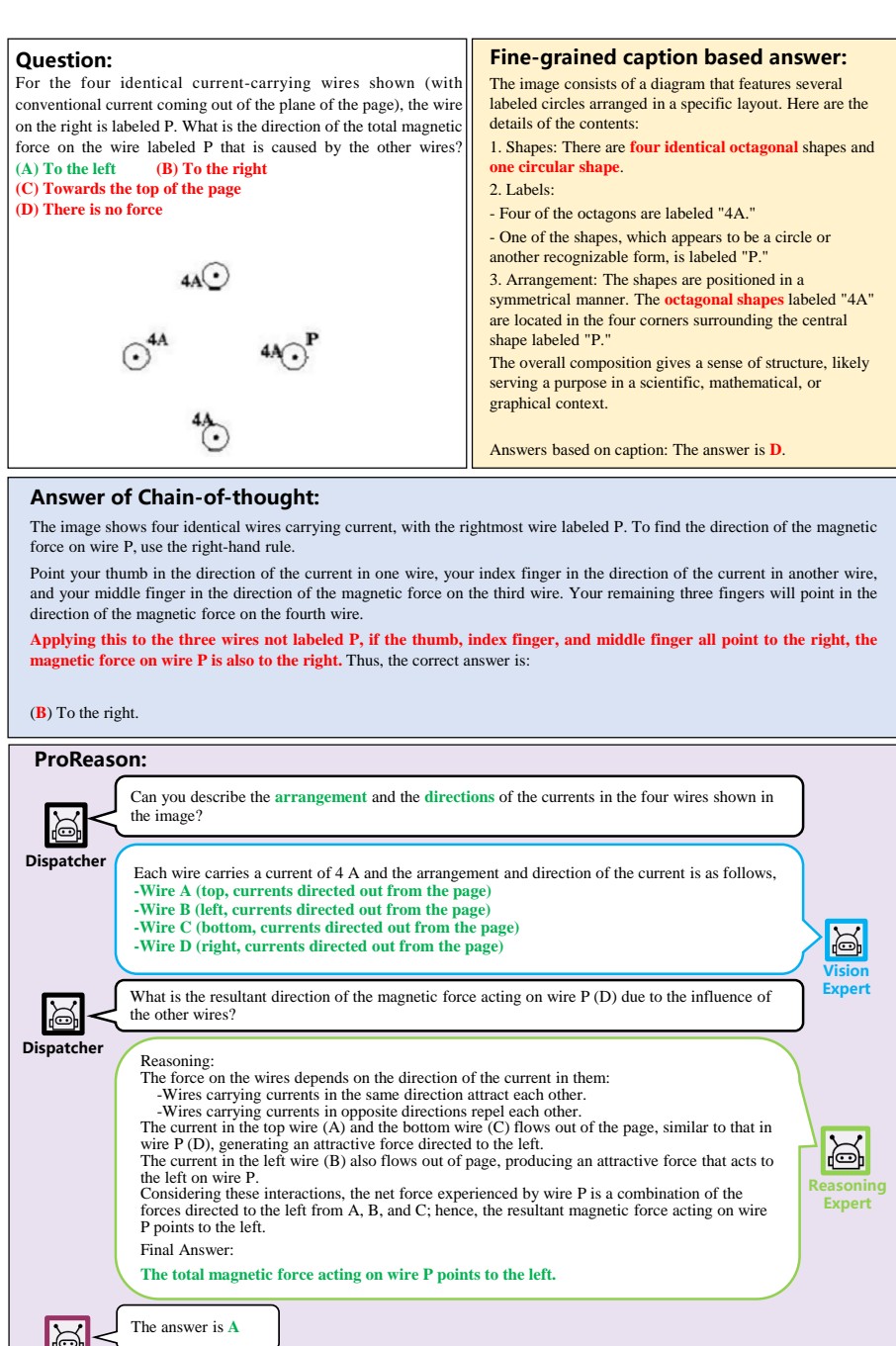

Figure 2: An example with three reasoning frameworks: fine-grained caption, chain-of-thought, and PROREASON. PROREASON enables LVLMs to proactively acquire necessary information in a question-oriented manner, and predicts answers based on the collected information. Apparently, PROREASON is superior to previous methods, which often describe question-irrelevant visual details, or overlook informative elements. Green indicates correct information or conclusions, while red signifies incorrect ones.

**Example case**

**Question:**
**For the four identical current-carrying wires shown (with conventional current coming out of the plane of the page), the wire on the right is labeled P. What is the direction of the total magnetic force on the wire labeled P that is caused by the other wires?**
**(A) To the left (B) To the right (C) Towards the top of the page (D) There is no force**

**[Dispatcher]:**
'Thought':'To determine the direction of the total magnetic force on wire P caused by the other wires, I need to visualize the arrangement and the effect of the magnetic fields created by the other wires, as well as the direction of the conventional current in each wire.',
'EXPERT name':'Vision Expert',
'Question':'Can you describe the **arrangement** and the **directions** of the currents in the four wires shown in the image? '

**[Vision Expert]:**
Each wire carries a current of 4 A and the arrangement and direction of the current is as follows,
**-Wire A (top, currents directed out from the page)**
**-Wire B (left, currents directed out from the page)**
**-Wire C (bottom, currents directed out from the page)**
**-Wire D (right, currents directed out from the page)**

**[Referee]:**
Thought: The arrangement and the directions of the currents in the four wires shown in the image are as follows::
-Wire A (top, currents directed out from the page)
-Wire B (left, currents directed out from the page)
-Wire C (bottom, currents directed out from the page)
-Wire D (right, currents directed out from the page)
To determine the total magnetic force exerted on wire P by the other three wires, it is necessary to apply the right-hand rule for analysis. Therefore, with the current information provided, it is not possible to directly obtain the answer.
Answer: UNSOLVABLE

**[Dispatcher]:**
'Thought':'To determine the total magnetic force on wire P due to the three surrounding wires, I need to use the right-hand rule and the fact that all currents are flowing in the same direction. Each wire will exert a magnetic force on wire P that can be calculated from their configurations.',
'EXPERT name':'Reasoning Expert',
'Question': 'What is the resultant direction of the magnetic force acting on wire P (D) due to the influence of the other wires? '

**[Reasoning expert]:**
Reasoning:
The force on the wires depends on the direction of the current in them: -Wires carrying currents in the same direction attract each other. -Wires carrying currents in opposite directions repel each other.
The current in the top wire (A) and the bottom wire (C) flows out of the page, similar to that in wire P (D), generating an attractive force directed to the left.
The current in the left wire (B) also flows out of page, producing an attractive force that acts to the left on wire P.
Considering these interactions, the net force experienced by wire P is a combination of the forces directed to the left from A, B, and C; hence, the resultant magnetic force acting on wire P points to the left.
Final Answer: The total magnetic force acting on wire P points to the left.

**[Referee]:**
Thought: Based on the known information, the total magnetic force acting on wire P points to the left.
Answer: SOLVABLE

**[Summarizer]:**
Thought: Given that the total magnetic force acting on wire P points to the left, when comparing this answer with the choices:
(A) To the left (B) To the right (C) Towards the top of the page (D) There is no force
It is evident that the correct choice is **(B)**.
Answer:**B**

Figure 3: A complete reasoning process of PROREASON for the case shown in Figure 2.

**Examples of picture unrelated Chain-of-thought.**

The graph below shows the AD-AS diagram for Spain. All numbers are in billions. <image 1>
What is the size of real GDP in the short-run equilibrium?
(A) 500 (B) 450 (C) 400 (D) 600
We refer to Wikipedia articles on macroeconomics for help. The short-run equilibrium real GDP is equal to the intersection of the SRAS curve and the vertical axis. The SRAS curve is a straight line with slope 200, and the vertical axis is at y = 500. Therefore, the answer is (A).

Based on <image 1>, what's the most likely diagnosis?
(A) first degree atrioventricular block   (B) third degree atrioventricular block
(C) Second degree type II atrioventricular block   (D) atrial flutter
We refer to Wikipedia articles on medicine for help. The given ECG shows a junctional bradycardia with a rate of 40 beats per minute, a P-R interval of 0.12 seconds, and a Q-T interval of 0.39 seconds. The PR interval is prolonged, which is consistent with second-degree type II atrioventricular block. Therefore, the most likely diagnosis is (C) Second degree type II atrioventricular block.

Figure 4: Additional examples of images-unrelated Chain-of-Thought reasoning.

**Examples of question unrelated caption.**

**Question:** How much do I have to pay according to the bill
**Caption:** The image shows an order details screen from a ride-hailing app, displaying a map of a city with two markers indicating the pickup location (green) and drop-off location (red). The total fare is calculated based on the base fare, mileage fee, and duration fee, with a discount applied resulting in the final amount paid. The image also includes a table summarizing the order details, including the base fare, mileage fee, duration fee, total cost, discount, and final amount paid. The image provides a clear overview of the ride details and pricing information for the user.

**Question:** When a negatively charged rod is brought near, but does not touch, the initially uncharged electroscope shown above, the leaves spring apart (I). When the electroscope is then touched with a finger, the leaves collapse (II). When next the finger and finally the rod are removed, the leaves spring apart a second time (III). The charge on the leaves is
(A) positive in both I and III   (B) negative in both I and III
(C) positive in I, negative in III  (D) negative in I, positive in III
**Caption:** The image shows three diagrams depicting the same hand holding a stick, but with different perspectives. In the first diagram, the hand is seen from above, holding the stick vertically. In the second diagram, the hand is seen from the side, holding the stick horizontally. In the third diagram, the hand is seen from the front, holding the stick at a diagonal angle. The diagrams show how perspective can affect our perception of objects and their orientation in space.

Figure 5: Additional examples of question-unrelated passive image captions.

A.2   ANALYSIS OF BAD CASES

In this section, we analyze the typical errors made by `GPT-4o-mini`-driven PROREASON, exemplified as Figure 6 and 7, for further understanding and potential research directions.

**Cumulative Errors.**   As illustrated in Figure 6, the vision expert mistakenly perceives the clock as 6:25, which misguides the reasoning of subsequent agents and ultimately leads to an incorrect conclusion. More broadly, similar misperceptions occur frequently in errors made by PROREASON. This indicates that, with the assistance of LLMs, PROREASON has effectively addressed the reasoning deficiencies in multi-modal tasks, while the vision expert plays a significant role for further improvement of multi-modal capabilities.

**Contradictory Information among Agents.**   Considering that multiple agents are engaged in the answering process, we try to find instances where contradictory information is provided by different agents, especially the vision expert and reasoning expert. However, as shown in Figure 6 and 7, when one agent (e.g., the vision expert) makes an error and the referee even hints at a possible mistake, other agents (e.g., the reasoning expert) tend to adhere to the available information instead of questioning it. This tendency results in a failure to find cases with contradictory information, and also highlights the importance of a reflection mechanism (Ji et al., 2023) in agent collaboration, which is left for future exploration.

**Bad case**

**Question: What time does the clock show?**
**Choices: (A) 1:30 (B) 6:30 (C) 5:30 (D) 4:30 (E) 12:30**
**[Dispatcher]:**
'Thought':'I need to determine what time the clock shows, which involves interpreting the image rather than complex reasoning.',
'EXPERT name':'Vision Expert',
'Question':'What time does the clock show in the image?'
**[Vision Expert]:**
The clock shows 6:25. The minute hand is pointing at the 5, indicating 25 minutes past the hour, and the hour hand is slightly past the 6.
**[Referee]:**
Thought: Given that the clock's display of 6:25 does not correspond with any of the answers provided, it can therefore be concluded that this question cannot be resolved with the available information as none of the options represent the correct time the clock shows.
Answer: UNSOLVABLE
**[Dispatcher]:**
'Thought': 'The clock shows that the hour hand is slightly past the 6 and the minute hand is pointing at the 5, which indicates that the time is 6:25. However, I need to choose the closest matching answer from the provided options.',
'EXPERT name': 'Reasoning Expert',
'Question': 'Given that the time shows 6:25, which of the provided answer choices (A: 1:30, B: 6:30, C: 5:30, D: 4:30, E: 12:30) is the closest to this time?'
**[Reasoning expert]:**
Reasoning:
(A) 1:30 - This is 4 hours and 5 minutes away from 6:25.
(B) 6:30 - This is 5 minutes away from 6:25.
(C) 5:30 - This is 55 minutes away from 6:25.
(D) 4:30 - This is 1 hour and 55 minutes away from 6:25.
(E) 12:30 - This is 5 hours and 55 minutes away from 6:25.
Now, the option that is the closest to the current time of 6:25 is: 6:30
Final Answer: (B) 6:30, which is only 5 minutes away.
**[Referee]:**
Thought: The clock currently shows 6:25. The closest time to 6:25 among the listed choices is (B) 6:30, as it is only 5 minutes away. Thus, this question can be resolved with the known information.
Answer: SOLVABLE
**[Summarizer]:**
Thought: The clock currently shows 6:25, meaning the hour hand is just above the 6 and the minute hand points at the 5, indicating that it is 25 minutes past 6. The closest time to 6:25 among the listed choices is (B) 6:30, as it is only 5 minutes away.
Answer:B

Figure 6: A typical mistake made by PROREASON. The vision expert incorrectly identifies 4:30 as 6:25, leading the other agents to base their judgments on this erroneous information, and ultimately resulting in the wrong conclusion.

**Bad case**

**Question:**
如图，在5×4的正方形网格中，每个小正方形的边长都是1，△ABC的顶点都在这些小正方形的顶点上，则 tan∠BAC的值为（）
As shown in the figure, in a 5×4 square grid where each small square has a side length of 1, the vertices of triangle ABC are located at the vertices of these small squares, so the value of tan∠BAC is ().

**Choices:** (A) $\frac{4}{3}$ (B) 0.75 (C) 0.6 (D) 0.8

**[Dispatcher]:**
'Thought':'To determine the value of tan∠BAC, I need to know the coordinates of points A, B, and C on the 5×4 grid. This information is represented in the image, and since I cannot see it, I should consult the Vision Expert to get the coordinates of these points.',
'EXPERT name':'Vision Expert',
'Question':'What are the coordinates of points A, B, and C in the 5×4 grid?'

**[Vision Expert]:**
In a 5×4 grid, we can assign coordinates to the points based on their positions:
- Point A is located at the top right corner of the grid, at coordinates **(4, 3)**.
- Point B is situated at the bottom left corner, at coordinates **(0, 0)**.
- Point C is at the bottom right corner of the grid, at coordinates **(4, 0)**.

**[Referee]:**
Thought: While the coordinates of points A, B, and C are known, the value of tan∠BAC remains undetermined, preventing the resolution of the problem.
Answer: UNSOLVABLE

**[Dispatcher]:**
'Thought':'Given the coordinates of A (4, 3), B (0, 0), and C (4, 0), the value of tan∠BAC still needs to be calculated.',
'EXPERT name':'Reasoning Expert ',
'Question':'Given the coordinates of A (4, 3), B (0, 0), and C (4, 0), calculate the value of tan∠BAC. '

**[Reasoning expert]:**
Reasoning:
To find the value of the tangent of angle BAC, we start by figuring out the slopes of the lines AB and AC in the triangle made up of points A, B, and C.
First, let's look at the coordinates of the points:
    Point A is located at (4, 3).
    Point B is located at (0, 0).
    Point C is located at (4, 0).
Next, we figure out the slope of line segment AB:
    The slope of AB, denoted as m_AB, can be found by dividing the difference in the y-coordinates by the difference in the x-coordinates, which gives us (0 - 3) / (0 - 4) = (-3) / (-4) = 3/4.
Then, we determine the slope of line segment AC:
    For AC, when we try to calculate its slope using the same method, we end up with (0 - 3) / (4 - 4), which results in division by zero. This tells us that the line AC is vertical and does not have a defined slope.
Given that AC is a vertical line and AB has a slope of 3/4, we can use this information to find the tangent of angle BAC. In a right triangle, the tangent of an angle is the length of the opposite side divided by the length of the adjacent side.
For angle BAC:
    Line AB rises 3 units vertically (from point B at (0, 0) to point A at (4, 3)) and runs 4 units horizontally (from point B at (0, 0) to point C at (4, 0)).
    Therefore, the tangent of angle BAC is the ratio of these two lengths: 3 divided by 4, which equals 0.75.
Final Answer: The tangent of angle BAC is 0.75.

**[Referee]:**
Thought: Based on the known information, the value of tan∠BAC is 0.75.
Answer: SOLVABLE

**[Summarizer]:**
Thought: Given that ∠BAC=0.75, when comparing this value with the answer choices:
(A) **4/3** (B) 0.75 (C) 0.6 (D) 0.8
It is evident that the correct choice is (B).
Answer:**B**

Figure 7: A typical mistake made by PROREASON. The vision expert incorrectly provided the coordinates of the three vertices of the triangle, leading the other agents to base their judgments on this erroneous information, and ultimately resulting in the wrong conclusion.

## A.3 PROMPTS

**Prompt templates of image Relevance Score .**

I answered a question based on what was in a picture, and here is the standard answer:
{standard answer}
Here is my answer:
{answer}
Standard answer effectively utilizes key information from the images, providing detailed and question-oriented image descriptions.
Based on the standard answer, please evaluate the relevance of my answer to the content of the image, on a scale of 1 to 5.

Please base your response on the following format:
Assessment process: analyze and assess here.
Final answer: one of ['1', '2', '3', '4','5']

**Prompt templates of caption effectiveness evaluation.**

I answered a question based on what was in a picture, and here is the question:
{question}
Here is the caption of the picture:
{caption}
And here is the standard answer:
{standard answer}
Standard answer effectively utilizes key information from the images, providing detailed and question-oriented image descriptions.
Based on the standard answer, please evaluate:
1. The level of detail in the caption.
2. The relevance of the caption to the question.
3. The extent to which the caption includes information used in the standard answer.
On a scale of 1 to 5.

Please base your response on the following format:
Assessment process: analyze and assess here.
Final answer:
The level of detail in the caption: one of ['1', '2', '3', '4','5']
The relevance of the caption to the question: one of ['1', '2', '3', '4','5']
The extent to which the caption includes information used in the standard answer: one of ['1', '2', '3', '4','5']

Figure 8: Prompt templates of Relevance Score and caption effectiveness evaluation.

**Prompt template of Chain-of-Thought.**

Please solve the following question with step-by-step reasoning: {question}

**Prompt template of fine-grained image captions generation.**

Please describe the contents of this image in detail: {image}

**Prompt template of Compositional Chain-of-Thought (CCoT).**

For the provided image and its associated question, generate only a scene graph in JSON format that includes the following:
1. Objects that are relevant to answering the question
2. Object attributes that are relevant to answering the question
3. Object relationships that are relevant to answering the question

**Prompt template of ReAct.**

Answer the following questions as best you can. You have access to the following tools:
image_description_tool:
Call this tool to interact with the Image Description Tool API.
Utilize this tool when you require insight into the components of an image, such as identifying objects or reading text within it.
Parameters:
[{'name': 'image_description_query',
'description': 'The input for this tool must be a question in string format. For example: The input could be, "What items are in this picture?"',
'required': True,
'schema': {'type': 'string'}}]
Format the arguments as a JSON object.
computational_tool:
Call this tool to interact with the computational tool API.
Use this tool when you need to conduct reasoning, such as calculating the current in a device with a voltage of 4 volts across and a resistance of 10 ohms, and similar scenarios.
Parameters:
[{'name': 'computational_query',
'description': 'The input for this tool must be a problem that requires calculation and reasoning. For example: The input could be, "What is the acceleration produced by a force of 10 Newtons acting on a 1-kilogram object?"',
'required': True,
'schema': {'type': 'string'}}]
Format the arguments as a JSON object.

Use the following format:
Question: the input question you must answer
Thought: you should always think 'step by step' about what to do
Action: the action to take, should be one of [image_description,computational_tool]
Action Input: the input to the action
Observation: the result of the action
... (this Thought/Action/Action Input/Observation can be repeated zero or more times)
Thought: I now know the final answer
Final Answer: the final answer to the original input question

Begin!

Figure 9: Prompt templates of Chain-of-Thought, fine-grained image captions generation, Compositional Chain-of-Thought (CCoT), and ReAct.

**Prompt template of Dispatcher.**

You currently need to address the following question:
{question}
The information you need is in an image, but you can't see the image right now.
At the same time, you're not capable of complex reasoning.

However, you can can consult the following two EXPERTs for help:
1. Vision Expert: You can ask him for information in the picture, for example, you could ask him, "What color is the bird in the picture?"
2. Reasoning Expert: You can ask him to get the results of complex reasoning, e.g. you can ask him, "What is the acceleration produced by a 1N force applied to a 1KG object?"

To solve this problem, which EXPERT do you think you should consult now?

Use the following format:
{
'Thought': 'analyze the problem here.',
'EXPERT name':'The name of the EXPERT you choose should be one of Vision Expert and Reasoning Expert',
'Question':'Questions you want to ask the EXPERT'
}

The last expert you chose was:
{last expert}
And the information you know currently is as follows:
{memory}

Figure 10: Prompt templates of Dispatcher.

**Prompt template of Vision expert.**

Please answer the following question in detail: {question}

**Prompt template of Reasoning expert.**

The following is the available information:
{memory}
Please solve the following problems step by step:
{question}
Use the following format:
Reasoning: Perform a step-by-step process of reasoning to solve a problem.
Final Answer: The final answer you get when you have finished reasoning.

**Prompt template of Referee.**

My current question that needs to be addressed is:
{question}
The following is the known information:
{memory}
Return SOLVABLE if you think question can be resolved with known information. Otherwise return UNSOLVABLE.
Use the following format:
Thought: Conduct an analysis before you give me an answer.
Answer: the action to take, should be one of ['SOLVABLE', 'UNSOLVABLE']

**Prompt template of Summarizer.**

My current question that needs to be addressed is:
{question}
The following is the known information:
{memory}
Please solve the question using the following format:
Thought: Conduct a step-by-step analysis before you give me an answer.
Answer: The final answer you get when you have finished analysis.

Figure 11: Prompt templates of Vision Expert, Reasoning Expert, Referee, and Summarizer.

## A.4 SUPPLEMENTARY RESULTS

Table 8: Average token consumption of PROREASON and ReAct with `GPT-4o-mini` model on the MME and MathVista benchmarks.

| Method | MME | | MathVista | |
|---|---|---|---|---|
| | Input | Output | Input | Output |
| PROREASON | 1286.8 | 327.2 | 2238.6 | 788.6 |
| ReACT | 1645.0 | 197.0 | 3092.8 | 845.1 |

Table 9: Performance of multiple approaches with two base models across four visual reasoning benchmarks. Different from Table 4, MME scores are calculated with the official method here.

| Model | Method | Dataset | | | |
|---|---|---|---|---|---|
| | | MME | MMMU | MathVista | HallusionBench |
| Llama3-LLaVA-NeXT-8B | Direct | 359 | 41.8 | 37.1 | 45.8 |
| | VDGD | 374 | 42.3 | 36.1 | 44.2 |
| | CCoT | 354 | 40.5 | 36.8 | 37.4 |
| | CoT | 258 | 41.5 | 35.9 | 43.1 |
| | ReAct | 385 | 46.7 | 31.7 | 43.6 |
| | PROREASON | 416 | 52.5 | 38.8 | 50.9 |
| GPT-4o-mini | Direct | 552 | 48.4 | 53.0 | 56.0 |
| | VDGD | 584 | 51.4 | 51.2 | 52.4 |
| | CCoT | 535 | 54.2 | 53.6 | 56.7 |
| | CoT | 729 | 58.5 | 53.8 | 56.3 |
| | ReAct | 712 | 54.8 | 49.3 | 51.1 |
| | PROREASON | 779 | 61.6 | 54.9 | 58.9 |

Table 10: Performance of `Llama3-LLaVA-NeXT-8B` Vision Expert, assisted by different text-only LLMs, on various benchmarks under ReAct and PROREASON frameworks. Different from Table 5, MME scores are calculated with the official method here.

| Vision Expert | Textual Sub-Agents | Method | Dataset | | | |
|---|---|---|---|---|---|---|
| | | | MME | MMMU | MathVista | Hallu. |
| Llama3-LLaVA-NeXT-8B | Llama3-LLaVA-NeXT-8B | ReAct | 385 | 46.7 | 31.7 | 43.6 |
| | | PROREASON | 416 | 52.5 | 38.8 | 50.9 |
| | Qwen2.5-72B-Instruct | ReAct | 428 | 50.4 | 34.6 | 40.4 |
| | | PROREASON | 603 | 56.8 | 48.8 | 52.3 |
| | GPT-4o-mini | ReAct | 473 | 48.4 | 36.2 | 46.7 |
| | | PROREASON | 613 | 54.5 | 41.7 | 53.1 |

Table 11: Performance of PROREASON across five scenarios for sub-agent assessment on visual reasoning tasks. For each scenario, one sub-agent is replacing with `Llama3-LLaVA-NeXT-8B`, while the others are performed by `GPT-4o-mini`. Different from Table 6, MME scores are calculated with the official method here.

| Dataset | GPT-4o-mini | Llama3-LLaVA-NeXT-8B | | | | |
| --- | --- | --- | --- | --- | --- | --- |
| | | Dispatcher | Vision Expert | Reasoning Expert | Referee | Summarizer |
| MME | 779 | 746 | 613 | 743 | 762 | 641 |
| MMMU | 61.6 | 60.9 | 54.5 | 60.2 | 51.5 | 51.0 |

Table 12: Performance of PROREASON with different configurations for the relative importance assessment between visual understanding and textual reasoning capabilities on visual reasoning tasks. Different from Table 7, MME scores are calculated with the official method here.

| Dataset | GPT-4o-mini | | Llama3-LLaVA-NeXT-8B | | |
| --- | --- | --- | --- | --- | --- |
| | Textual Sub-Agents | Vision Expert | All Sub-Agents | COT | Direct |
| MME | 613 | 512 | 416 | 286 | 359 |
| MMMU | 54.5 | 53.4 | 52.5 | 41.5 | 41.8 |

