# OpenReview forum: "ProReason: Multi-Modal Proactive Reasoning with Decoupled Eyesight and Wisdom"
_ICLR.cc/2025/Conference — Submitted to ICLR 2025_

### Official Review · Reviewer_BRG2 · 2024-11-02

**Soundness:** 3
**Presentation:** 3
**Contribution:** 2
**Rating:** 6
**Confidence:** 5

**Summary:**

This paper introduces PROREASON, a novel multi-modal reasoning framework that decouples visual perception and textual reasoning capabilities in large vision-language models (LVLMs). The framework features a multi-agent system that proactively collects visual information based on questions and performs reasoning through separate specialized components. The authors demonstrate that PROREASON outperforms existing approaches across multiple benchmarks, with improvements up to 13.2% on standard metrics.

**Strengths:**

I majorly conclude there are two strengths in this paper.

The paper presents a compelling approach to separating visual perception from textual reasoning, addressing a fundamental limitation of current LVLMs. This decomposition allows for more effective handling of each capability and enables the integration of specialized models for different aspects of the task.

The framework's ability to seamlessly integrate existing LLMs for improved reasoning capabilities is particularly valuable, as it allows organizations to leverage their existing investments in language models while enhancing multi-modal capabilities.

**Weaknesses:**

There are two main weakness about this paper.

1. The paper doesn't thoroughly discuss how the framework handles cases where the Vision Expert and Reasoning Expert disagree or provide conflicting information. Suggestion: Add a section analyzing failure cases and how the framework handles conflicting information between agents.

2. The evaluation focuses on specific benchmark tasks, but doesn't extensively explore how the framework scales with increasing complexity of visual scenes or reasoning requirements. Suggestion: Include experiments with varying levels of visual and reasoning complexity to demonstrate scalability.

**Questions:**

1. How does the framework handle cases where the required visual information is implicit or requires complex inference from multiple parts of the image? For example, in scenarios where understanding spatial relationships or temporal sequences is crucial?

2. Could the authors elaborate on how the framework might be extended to handle multiple images or video inputs, where temporal reasoning and cross-frame information integration become important?

---

> ### Author Response · Authors · 2024-11-25
> **Response to reviewer**
>
> Thanks for your great dedication and thoughtful comments on our work! We will clarify them one by one with our greatest efforts below. Hope that we can address your concerns!
>
> **Weakness 1: Lack of Failure Cases & Conflicting Information Analysis among Agents.**
> - Thanks for your valuable suggestion! In our updated submission, we **supplemented bad case analysis in Appendix A.2.** If interested, you can check this section for details now.
> - As suggested by you, we discussed the contradictory information among agents. Specifically, we tried to identify instances where contradictory information was provided by different agents, particularly the vision and reasoning experts. However, as shown in Figure 5 and 6 in our updated submission, when one agent (e.g., the vision expert) makes an error and the referee even hints at a possible mistake, other agents (e.g., the reasoning expert) tend to adhere to the available information instead of questioning it. This tendency results in a failure to find cases with contradictory information, and also highlights the importance of a reflection mechanism [1] in agent collaboration, which is left for future exploration.

---

> ### Author Response · Authors · 2024-11-25
> **Response to reviewer (2/n)**
>
> **Weakness 2: Scaling with Increased Complexity of Visual Scenes or Reasoning Requirements.**
> - We fully understand your concerns regarding the scalability of ProReason. Among the benchmarks included in our paper, MMMU exhibits higher complexity in both visual and reasoning aspects compared to MME. For more comprehensive comparisons, we test on an additional VQA dataset, A-OKVQA [2]. The visual complexity of A-OKVQA is comparable to MME, but its reasoning requirements are less demanding.
> - The table below demonstrates that ProReason achieves the highest performance across all three benchmarks. As the visual and reasoning complexity increases, ProReason outperforms "Direct" method by 2.7%, 11.2%, and 13.2% on A-OKVQA, MME, and MMMU, respectively. This indicates that ProReason is particularly effective on datasets with greater complexity, due to its decoupled capabitilies and LLM-assisted reasoning. Thanks for your advice! We will include the results and analysis in our final paper.
>
> | Dataset    |        |      | GPT-4o-mini |      |       |             |
> | ---------- | ------ | ---- | ----------- | ---- | ----- | ----------- |
> |            | Direct | VDGD | CCoT        | CoT  | ReAct | ProReason   |
> | A-OKVQA[4] | 78.6   | 79.4 | 79.2        | 80.9 | 80.6  | 81.3(+2.7)  |
> | MME        | 79.2   | 82.3 | 80.8        | 87.8 | 87.3  | 90.4(+11.2) |
> | MMMU       | 48.4   | 51.4 | 54.2        | 58.5 | 54.8  | 61.6(+13.2) |

---

> ### Author Response · Authors · 2024-11-25
> **Response to reviewer (3/n)**
>
> **Question 1: Handling Implicit Visual Information and Complex Inference.**
> - It is noteworthy that **the vision expert is not limited to generating only the most atomic information** (e.g., color or size of individual objects), because imposing such a restriction would disrupt the relationships among various objects and impede the integration of information. Instead, the vision expert is permitted to provide spatial or temporal information as needed (e.g., relative size or position of objects). This information will then be enhanced by the reasoning expert to yield more comprehensive insights (e.g., temporal relationships among different parts of a flowchart on a high level). Consequently, **ProReason should be inherently capable of handling spatial and temporal relationships based on this composite information, akin to LVLMs.** Hope that this could address your concerns!

---

> ### Author Response · Authors · 2024-11-25
> **Response to reviewer (4/n)**
>
> **Question 2: Extending Framework for Multi-Image and Video Inputs.**
> - Overall, **ProReason should be easily extended to multi-image or video tasks with the generalizability of this framework.** Specifically, visual perception is primarily handled by the vision expert. For multi-image or video tasks, we can utilize a multi-image or video model as the vision expert to describe multi-image or multi-frame information in a sequential order, which is then organized in the memory. Subsequently, other agents can interact within ProReason to generate better output with the powerful reasoning abilities of LLMs.
> - Alternatively, we can still employ a single-image vision expert to describe key images individually, and then organize the information in the memory. Afterwards, the dispatcher can request specific image information from the vision expert or analytical information (e.g., scene comprehension and inferential insights) from the reasoning expert, based on the requirements of the task. This workflow still adheres to the ProReason framework, but allows the single-image vision expert to perform multi-image and video tasks. We believe that **enhancing multi-image and video tasks with the powerful reasoning capabilities of LLMs is a promising research direction, which deserves much more exploration.**
>
> Thank you again for your dedication and constructive feedback to our work! Sincerely, we hope that these clarifications can fully address your concerns!
>
> **References**
>
> [1] Ji, Z., Yu, T., Xu, Y., Lee, N., Ishii, E., & Fung, P. (2023, December). Towards mitigating LLM hallucination via self reflection. In Findings of the Association for Computational Linguistics: EMNLP 2023 (pp. 1827-1843).
>
> [2] Schwenk D, Khandelwal A, Clark C, et al. A-okvqa: Visual qa using world knowledge. In: ECCV, 2022.

---

> ### Author Response · Authors · 2024-11-28
> **Response to reviewer BRG2**
>
> Dear Reviewer BRG2,
>
> We fully understand that you may have been very busy recently!
>
> Given that the rebuttal deadline is approaching, we genuinely hope that our clarifications have addressed your concerns and will encourage you to reconsider the scores. If you have any further questions, please do not hesitate to discuss them with us！
>
> Looking forward to your further feedback!
>
> Sincerely,
>
> Authors

---

### Official Review · Reviewer_zwCa · 2024-11-05

**Soundness:** 2
**Presentation:** 2
**Contribution:** 2
**Rating:** 6
**Confidence:** 5

**Summary:**

Briefly, this paper presents a multi-step multi-modal reasoning framework to overcome the limitation of LVLMs for visual reasoning tasks by introducing visual perception (i.e., eyesight) and textual reasoning (i.e., wisdom). The experimental results are strong. Besides,  the authors demonstrate the drawbacks of existing solutions (i.e., insufficient yet irrelevant visual descriptions and limited multi-modal capacities).

**Strengths:**

+ The proposed method presents an effective way to extract the necessary and sufficient visual details from images for the further multi-model reasoning step.
+ Extensive and comprehensive experiments demonstrate the superiority of the proposed method.

**Weaknesses:**

1) Sub-optimal Design and Assumption. In Sect. 2.2, the authors argue that a detailed caption of the given image cannot provide sufficient and relevant information for Visual Reasoning (VR). However, some works (e.g., [s1]) concentrate on optimizing the caption. It might be more reasonable to include an optimized caption for the proposed Action step rather than use Q-I as input.

2) Insufficient Experimental Evaluation.
   + Why isn't GPT-4V included for comparison? It is somewhat difficult to fairly evaluate the proposed method on some benchmarks (e.g., HallusionBench) that have already reported the performance of GPT-4V.
   + Since the proposed method includes many steps during the inference, it might suffer from accumulated errors, e.g., wrong answers from the action step. The ablation study should include an extra experiment with a detailed discussion.
   + Missing the experiments on the traditional open-domain VQA benchmark (e.g., GQA).  It seems that the time complexity of the proposed method is high for some simple VQA tasks.

3) Due to the poor presentation of the Summary step, the proposed method might only works well on the VR task with multiple choices provided.

[s1] Hu et al., PROMPTCAP: Prompt-Guided Image Captioning for VQA with GPT-3. In CVPR 2023.

**Questions:**

Please refer to the weaknesses section.

---

> ### Author Response · Authors · 2024-11-21
> **Response to reviewer**
>
> Thanks for your valuable reviews on our work, and the provided paper [1]! We have read the paper carefully. Briefly, this paper trains a question-aware captioning model (i.e., PromptCap) to generate better captions. Below, we try to answer your questions one by one with our greatest efforts. Hope that we could address your concerns!
>
> **Weakness 1: Sub-optimal Design and Assumption.**
> - **Sub-optimal Assumption:** We acknowledge that to some extent, PromptCap mitigates the problems of  irrelevant and insufficient captions with a finetuned captioning model. However, **PromptCap is far from a "perfect" solution.** Imagine that with a "perfect" captioning model and powerful reasoning capabilities of LLMs, most multi-modal problems should have been solved. The fact is that there is lots of work that needs to be done to improve the multi-modal capacities. Hence, we think that **it is still reasonable to claim, but in a more precise way, that the multi-modal capacities still suffer from limited visual perception ability (i.e., irrelevant and insufficient captioning).**  We will also adjust the corresponding claims in our camera-ready submission later.
> - **Sub-optimal Design:**
>   - Honestly, we are not that sure if PromptCap can further improve the Action step of ProReason. Specifically, **PromptCap is a finetuned captioning model, playing a similar function to the Vision Expert of ProReason.** However, ProReason adoptes LLaVA or GPT-4o-mini as the Vision Expert. With extensive instruction tuning, **both models can be regarded as powerful captioning models** when provided with appropriate prompts (including the questions). Without thorough comparisons, we cannot assert if they can outperform PromptCap, but it seems that we can expect this with the rapid advancements of LLaVA and GPT models.
>   - **The main contribution of ProReason lies in decoupled perception and reasoning, which further enables improved LLM-assisted multi-modal reasoning.** This will potentially impact future research broadly. For example, given an existing large vision-language model (LVLM), we can use LLM to improve its multi-modal capabilities, producing better reasoning paths. This can be further used to train the LVLM. This loop, like self-improvement, may continuously improve the capabilities of the LVLM.
>   - In comparison, **how to generate better captions is also important, but not our main contribution.** Due to extensive optimization tricks in current research, it is also unrealistic to include all of them in a single paper, but we still **thank you for your insightful suggestions and great efforts on our work!**

---

> ### Author Response · Authors · 2024-11-21
> **Response to reviewer (2/n)**
>
> **Weakness 2: Insufficient Experimental Evaluation**
> 1. **Not included GPT-4V & Concern about evaluation fairness.**
>   - We fully understand your concerns regarding the fairness of evaluation. To address your concerns, we would like to clarify the following two points:
>     - Firstly, we have **provided all the prompts in Appendix A.3** for readers' reference, and will **open-source all the code** to ensure the reproducibility of our results upon publication.
>     - Additionally, **GPT-4o-mini is a better choice with balanced costs and performance.** In detail, compared to GPT-4V, GPT-4o-mini is newer, and exhibits better multi-modal performance. The table below presents a performance comparison of GPT-4V and GPT-4o-mini across three benchmarks, with the publicly reported results of GPT-4V [2, 3, 4]. Apparently, GPT-4o-mini outperforms GPT-4V, demonstrating the suitability of GPT-4o-mini to reflect the latest model capabilities. Besides, GPT-4o-mini offers ten times lower API pricing compared to both GPT-4o and GPT-4V, making it much more affordable for our experiments.
> | Dataset | GPT-4V | GPT-4o-mini |
> |-----|-----|-----|
> | MMMU [2] | 56.8 | 58.5 |
> | MME Cognition (Visual Reasoning) [3] | 517 | 729 |
> | Mathvista [4] | 49.9 | 53.8 |
>
> 2. **Discussion of accumulated errors.**
>   - We also agree that examining accumulated errors is certainly valuable. We checked existing multi-step reasoning frameworks, such as ReAct [5] and ToT [6], so that we can find solutions to resolve the difficulty in measuring accumulated errors. However, we find that they may also not find good solutions, and do not discuss the impact of accumulated errors. We sincerely welcome any discussions about the accumulated errors, especially the measuring methods!
>   - For your reference, we **supplement the qualitative analysis of cumulative errors made by ProReason in Appendix A.2.** As illustrated in Figure 6 of our latest submission, the vision expert mistakenly perceives the clock as 6:25, which misguides the reasoning of subsequent agents and ultimately leads to an incorrect conclusion. More broadly, similar misperceptions occur frequently in errors made by ProReason. This indicates that, with the assistance of LLMs, ProReason has effectively addressed the reasoning deficiencies in multi-modal tasks, while the vision expert plays a significant role for further improvement of multi-modal capabilities.
>   - Besides, **ProReason demonstrates significantly improved overall performance, suggesting fewer errors made by ProReason.**
> 3. **Missing VQA benchmark.**
>   - As mentioned above and also in our paper, ProReason targets improving the multi-modal reasoning abilities of a LVLM, which involves perception and reasoning simultaneously. In comparison, a traditional VQA task mainly requires visual perception, which is not located in our scope, and we'd like to attribute it to the improvement of vision encoder or the proposal of new multi-modal architectures. Besides, we can expect that ProReason performs similarly to its vision expert (i.e., a single LVLM, responsible for vision perception).
>   - Moreover, we **conduct an additional evaluation on the A-OKVQA dataset** [7]. This benchmark is utilized in your provided paper [1] for knowledge-based VQA, which requires vision perception and commonsense reasoning. As shown in the table below, **ProReason still outperforms all the other baselines**, indicating the positive impact of ProReason on any form of reasoning-involved multi-modal tasks. Thanks for your advice! We will include these results in the paper.
>
> | Dataset |           |           | GPT-4o-mini|           |           |           |
> |-----------|----|---|---|----|-----|----|
> |          | Direct | VDGD | CCoT | CoT  | ReAct | ProReason |
> | A-OKVQA[7]| 78.6   | 79.4 | 79.2 | 80.9 | 80.6  | 81.3     |
>
> .

---

> ### Author Response · Authors · 2024-11-21
> **Response to reviewer (3/n)**
>
> **Weakness 3: Only work well when multiple choices are provided.**
> - It is worth noticing that **the evaluation benchmarks are not limited to a multiple-choice style.** For instance, most questions in HallusionBench do not provide multiple choices, and require models to answer them based on visual information. As shown in Table 5 in our paper, ProReason outperforms exceptionally well on this benchmark, showing its generalization on non-multiple-choice questions.
> - For your reference, we **additionally conduct comparative experiments** with GPT-4o-mini on the non-multiple-choice (free-form) questions in the MathVista benchmark, accounting to 46% of all the samples. As shown in the following table, ProReason attains the highest score in the non-multiple-choice subset, **highlighting its superiority on the non-multiple-choice style again.** This analysis will also be supplemented in our final paper.
>
> | Dataset |          |          | GPT-4o-mini|          |          |          |
> |-----------|----|---|---|----|-----|----|
> |          | Direct | VDGD | CCoT | CoT  | ReAct | ProReason |
> | Mathvista [4]| 41.7   | 42.8 | 42.6 | 43.2 | 38.7  | 44.9     |
>
>
>
> We sincerely welcome any further discussions with the hope of fully addressing your concerns, and you can reconsider our scores!
>
>
> **References**
>
> [1] Hu Y, Hua H, Yang Z, et al. Promptcap: Guided image captioning for vqa with gpt-3. In: CVPR, 2023.
>
> [2] Yue X, Ni Y, Zhang K, et al. Mmmu: Multi-discipline multimodal understanding benchmark. In: CVPR, 2024.
>
> [3] https://github.com/BradyFU/Awesome-Multimodal-Large-Language-Models/tree/Evaluation?tab=readme-ov-file#cognition
>
> [4] Lu P, Bansal H, Xia T, et al. Mathvista: Evaluating math reasoning in visual contexts. arXiv:2310.02255, 2023.
>
> [5] Yao S, Zhao J, Yu D, et al. React: Reasoning and acting in language models. arXiv:2210.03629, 2022.
>
> [6] Yao S, Yu D, Zhao J, et al. Tree of thoughts: Problem solving with llms. In: NeurIPS, 2024.
>
> [7] Schwenk D, Khandelwal A, Clark C, et al. A-okvqa: Visual qa using world knowledge. In: ECCV, 2022.

---

> ### Author Response · Authors · 2024-11-28
> **Response to reviewer zwCa**
>
> Dear Reviewer zwCa,
>
> We fully understand that you may have been very busy recently!
>
> Given that the rebuttal deadline is approaching, we genuinely hope that our clarifications have addressed your concerns and will encourage you to reconsider the scores. If you have any further questions, please do not hesitate to discuss them with us！
>
> Looking forward to your further feedback!
>
> Sincerely,
>
> Authors

---

> > ### Comment · Reviewer_zwCa · 2024-12-02
> >
> > I would like to thank the authors deeply for the rebuttal. Most of my concerns have been addressed. I will give a higher rating.

---

> ### Author Response · Authors · 2024-12-02
> **Response to reviewer zwCa**
>
> Dear Reviewer zwCa,
>
> We sincerely appreciate your thorough feedback and are grateful for your decision to raise the score. We welcome any further discussions to fully address your concerns!
>
> Sincerely,
>
> Authors

---

### Official Review · Reviewer_ZzMj · 2024-11-09

**Soundness:** 3
**Presentation:** 3
**Contribution:** 3
**Rating:** 6
**Confidence:** 4

**Summary:**

This paper discusses the performance degradation problem of Large Vision-language models (LVLMs), which tend to rely more on language knowledge than image information in visual reasoning tasks. The authors first analyze the limitations of current solutions, then introduce the ProReason framework, which incorporates multi-run proactive perception and decoupled vision-language capabilities. It allows integration of the existing LLMs to compensate for the reasoning deficits of LVLMs. In detail, ProReason contains 5 components, including a dispatcher, a vision expert, a reasoning expert, a referee, and a summarizer.  The dispatcher selects a vision expert or a reasoning expert. After several iterations, a referee will decide if the information is enough for a summarizer to answer the question. To demonstrate the effectiveness of the proposed framework, experiments on several benchmarks show that ProReason outperforms existing multi-step reasoning frameworks and passive peer methods.

**Strengths:**

+ The paper provides a detailed analysis of the limitations in existing models, highlighting how current LVLMs tend to rely more on language information than on visual cues. This analysis points out issues such as insufficient and irrelevant visual descriptions and limited multi-modal capabilities.
+ The proposed ProReason framework can iteratively generate proactive perception and effectively decouple vision and language capabilities.
+ Extensive results on four benchmarks demonstrate ProReason's effectiveness. Additionally, ProReason’s design allows future integration with LLMs to enhance visual reasoning, showcasing the modulization and the potential for LVLM with LLM-assisted.

**Weaknesses:**

- The authors claim to 'decouple' multi-modal reasoning into visual perception and textual reasoning. However, they do not provide evaluations on key aspects such as the frequency with which the dispatcher selects the Vision Expert versus the Reasoning Expert, the content of the generated memory from each expert, and the relevance between the memory generated by the Vision Expert and the Reasoning Expert and standard answers.
- The authors compute relevance scores and evaluate caption effectiveness across different methods. However, since ProReason answers questions based on information stored in memory, calculating the relevance score between standard answers and the information within the memory could be an effective way to evaluate whether ProReason captures accurate information.
- In Figure 2, the example doesn’t fully clarify each component’s function, and the prompts differ from those in Figures 7 and 8, creating some inconsistency. Also, it would be great to show how memory is stored and given to each component.

**Questions:**

Questions are in the weaknesses. Here is the short summary:
1. Detailed evaluations on the Vision expert and Reasoning expert.
2. Calculating the relevance score between standard answers and the information stored in memory could enhance the evaluation of ProReason’s information accuracy.
3. Provide more details in the example for greater clarity.

---

> ### Author Response · Authors · 2024-11-25
> **Response to reviewer**
>
> We sincerely appreciate your insightful feedback and devoted time on our work! We will address each of your comments with our greatest efforts below, and hope that we can fully resolve your concerns!
>
> **Weakness 1: Detailed Evaluations on the Vision Expert and Reasoning Expert.**
>
> **1. Frequency of Vision Expert and Reasoning Expert.**
>
> Thanks for raising such a valuable question!
> - As listed in the table below, we evaluate the frequency of the Dispatcher choosing the Vision Expert or Reasoning Expert on both MME and MMMU benchmarks, with MMMU requiring higher visual and reasoning abilities. Specifically, compared to MME, the frequencies for both the Vision and Reasoning Experts are higher on the MMMU benchmark, aligning with their difficulty levels.  Together with the results in Table 4 of our submission, **ProReason can adaptatively increase the frequencies of experts, and provide consistent performance improvements (i.e., 11.2% and 13.2%).** Despite the lower frequency of the reasoning expert, the significant performance enhancement highlights the importance of LLM-assisted reasoning capabilities for reasoning-essential questions. Additionally, the frequency of the Vision expert exceeding 1 underscores the importance of referees, which controls the loop to call experts multiple times, alleviating the issue of insufficient information. We will supplement both the results and analysis to our final paper!
>
> | **Dataset** | **GPT-4o-mini**   |                      |
> | ----------- | ----------------- | -------------------- |
> |             | **Vision Expert** | **Reasoning Expert** |
> | MME         | 1.16              | 0.12                 |
> | MMMU        | 1.64              | 0.38                 |
>
> **2. Content Analysis of Vision Expert, Reasoning Expert and Standard Answers.**
> - Due to the overlap, we answer this question in "Weakness 2: Evaluating Relevance Score between Standard Answers and Memory". Please refer to the next paragraph for details.

---

> ### Author Response · Authors · 2024-11-25
> **Response to reviewer (2/n)**
>
> **Weakness 2: Evaluating Relevance Score between Standard Answers and Memory**
> - Thanks for your suggestion! Employing LLMs to evaluate information generated by various experts is indeed an effective approach. However, it is noteworthy that most of current benchmarks provide only the final answers rather than detailed golden reasoning process. In Sec. 2.1, we adopt the reasoning chains generated by GPT-4o-mini (CoT) as the standard answers, because GPT-4o-mini with a score of 58.5 notably surpasses other used models on the MMMU benchmark. Nevertheless, since ProReason improves GPT-4o-mini's score to 61.6, the answers of GPT-4o-mini are no longer appropriate. We, instead, choose GPT-4o to produce the standard answers with a score of 69.1 on MMMU [1].
> - As shown in the table below, we evaluate the reasoning process of GPT-4o-mini (CoT), the Memory of ProReason (ProReason-Memory), and the reasoning process of Summarizer (ProReason-Summarizer), which generates the final answer of Proreason. This assessment focuses on three key metrics: the relevance to standard answers (RE ↑), the degree of redundant information (RI ↓), and the extent of missing information (MI↓), where arrows indicate the directions of improvement.
> - Specifically, compared to GPT-4o-mini (CoT), **ProReason-Summarizer produces more relevant answers with less redundancy and deficiency, aligning with its improved performance.** Compared to ProReason-Summarizer, ProReason-Memory exhibits the same RE,  higher RI and lower MI scores. This suggests that ProReason allows some redundancy to prevent information loss in memory, as the former typically leads to more serious consequences than the latter. Subsequently, Summarizer can leverage its powerful reasoning capabilities to select the most relevant memory. These results further demonstrate the details of ProReason, which will be also included in our final paper!
>
> -  These results further validate the effectiveness of ProReason, which we will include in the paper.
>
> | Dataset | Score | GPT-4o-mini |                  |                      |
> | ------- | ----- | ----------- | ---------------- | -------------------- |
> |         |       | CoT         | ProReason-Memory | ProReason-Summarizer |
> | MMMU    | RE ↑ | 4.67        | 4.83             | 4.83                 |
> |         | RI ↓ | 3.33        | 3.66             | 2.83                 |
> |         | MI ↓ | 1.40        | 1.17             | 1.33                 |

---

> ### Author Response · Authors · 2024-11-25
> **Response to reviewer (3/n)**
>
> Weakness 3: Provide more details in the example for greater clarity
> 1. Inconsistent and Unclear Components in Figure 2.
> - Thanks for your suggestion, and we feel so sorry for our confusing presentations! The purpose of Figure 2 mainly lies in the high-level comparisons among three reasoning frameworks: fine-grained caption, chain-of-thought, and ProReason. Considering that the lengthy answers, presenting the entire reasoning process would be too messy, hindering the comprehension of readers. Therefore, we simplify the complete reasoning process for better readability in Figure 2. **For your reference, we supplement the complete reasoning process for the case in Figure 3** with the same prompts as Figures 10 and 11 in our updated submission (i.e., Figures 7 and 8 in our original version).
> 2. Details of Memory.
> - The main contribution of ProReason lies in decoupled perception and reasoning, which further enables improved LLM-assisted multi-modal reasoning. Hence, we simply set the Meomry as a text string. At the start of the reasoning process, the Memory is an empty string; as the reasoning progresses, the Vision Expert and Reasoning Expert append contents to it. We find that this straightforward design is enough for improved performance of ProReason. We do not rule out the  possibility of better alternatives, which is left for further exploration when it hinders the performance.
>
> For Question 1, 2, and 3, please refer to our clarifications on Weekness 1, 2, and 3, respectively.
>
> Thank you again for your constructive comments and your great efforts on our work! They do help us rethink our method from different perspectives. Sincerely, we hope that these clarifications can address your concerns, and our scores can also be improved accordingly!
>
>
>
> **References**
>
> [1] https://mmmu-benchmark.github.io/#leaderboard

---

> ### Author Response · Authors · 2024-11-28
> **Response to reviewer ZzMj**
>
> Dear Reviewer ZzMj,
>
> We fully understand that you may have been very busy recently!
>
> Given that the rebuttal deadline is approaching, we genuinely hope that our clarifications have addressed your concerns and will encourage you to reconsider the scores. If you have any further questions, please do not hesitate to discuss them with us！
>
> Looking forward to your further feedback!
>
> Sincerely,
>
> Authors

---

> > ### Comment · Reviewer_ZzMj · 2024-11-30
> > **Response to authors**
> >
> > Thank you to the authors for the rebuttal. I agree that most current benchmarks provide only final answers rather than a detailed golden reasoning process. However, I am curious if the authors have considered the VCR [1] dataset before. Since the VCR dataset includes "Rationale" for each question, I wonder if this could be relevant to the concept of "Memory" in your work.
> >
> > [1] Zellers, R., Bisk, Y., Farhadi, A., Choi, Y. From recognition to cognition: Visual commonsense reasoning. CVPR, 2019.
> >
> > I will keep my rating as 6.

---

> > > ### Author Response · Authors · 2024-12-04
> > > **Response to reviewer ZzMj**
> > >
> > > Thanks for your valuable feedback and for providing the paper [1]. We have carefully read the paper. Briefly, it introduces the VCR dataset, which includes a "Rationale" for each question. Unfortunately, we feel so sorry that this is the last chance for us to discuss with you, but we are still trying analyzing on the VCR dataset.
> > >
> > > We fully understand your concerns regarding the suitability of using GPT-4o to generate the golden answers, which introduces some flaws into the analyze required by you. However, as detailed in our previous response, this is a straightforward and cost-effective way to approximately compare the answers of different methods, particularly given that GPT-4o produces the best performance. Similarly, this approximation is also adopted by [2].
> > >
> > > Thank you once again for your suggestions! Your feedback helps us a lot to polish our work, and we will continue the analysis on the VCR dataset for more precise analysis!
> > >
> > > References
> > >
> > > [1]Zellers, R., Bisk, Y., Farhadi, A., & Choi, Y. (2019). From recognition to cognition: Visual commonsense reasoning. In Proceedings of the IEEE/CVF conference on computer vision and pattern recognition (pp. 6720-6731).
> > >
> > > [2]Liu, H., Li, C., Wu, Q., & Lee, Y. J. (2024). Visual instruction tuning. Advances in neural information processing systems, 36.

---

### Meta-Review · Area_Chair_CNLE · 2024-12-20

**Metareview:**

Paper addresses degradation of the performance of VLMs that stem from them relying less on visual information and more on language priors. The proposed approach amounts to Chain-of-Reasoning with multiple agents, which, in-part decouple visual information gathering and reasoning to produce answers. Paper was reviewed by three expert reviewers and, post rebuttal, received 3 x marginally above the acceptance threshold ratings. Main concerns of reviewers stem from: (1) lack of explanation and motivation for individual components, (2) evaluation of decoupling the vision and language information in processing, (3) lack of experiments with GPT-4V, and (4) lack of more comprehensive evaluation that ensures that intermediate reasoning steps are valid. Authors have provided a comprehensive rebuttal that reviewers have generally found compelling with [zwCa] increasing the rating. However, no reviewer was willing to champion the paper and, unfortunately, [BRG2] did not engage in discussion.

AC has reviewed the reviews, rebuttal, discussion that followed and the paper itself. Generally, while overall performance of the model is indeed strong and impressive, AC, similar to reviewers, is concerned that internal reasoning may not in fact be consistent and using final performance as a proxy of this (as is suggested by authors in the rebuttal) is generally insufficient. The overall design is also similar to various recent approaches that either do CoT or use tools (e.g., VisualProgramming CVPR'23 or VisualScatchpad from NeurIPS'24). It is not clear which specific designs make the proposed framework more suited for in-depth reasoning in complex scenarios.

Given the borderline nature of the paper, this paper was also discussed at length between AC and the Senior AC. This discussion has converged to a consensus of Reject recommendation at this time. It was felt that novelty was limited and justification for performance could use improvements. As such, authors are encouraged to revise the paper and to resubmit it to a future venue.

**Additional Comments On Reviewer Discussion:**

Authors have provided a comprehensive rebuttal that reviewers have generally found compelling with [zwCa] increasing the rating. However, no reviewer was willing to champion the paper and, unfortunately, [BRG2] did not engage in discussion. Paper was also discussed at length between AC and Senior AC that has ultimately led to the final recommendation above.

---

### Decision · Program_Chairs · 2025-01-22

Reject